# Impact of early work start on mental health outcomes in older adults: A cross-sectional study from Ecuador

**Romina Domínguez**[1], **Marco Faytong-Haro**[1,2,3] *

1 Facultad de Ciencias de la Salud, Universidad Espíritu Santo, Samborondon, Ecuador, 2 School of International Studies, Universidad Espíritu Santo, Samborondon, Ecuador, 3 Ecuadorian Development Research Lab, Daule, Ecuador

* mfaytong@uees.edu.ec

**Data Availability Statement:** All data supporting the findings of this study are openly available for public access. Interested parties can download the dataset from the National Institute of Statistics and

## Abstract

### Objective

This study assesses the impact of the age at which individuals first begin working on the odds of developing mental health disorders among older adults in Ecuador.

### Methods

Data from the 2009 Survey of Health, Well-being, and Aging (SABE) encompassing 3093 elderly participants from mainland Ecuador were analyzed. We employed binary logistic regression to explore the association between the age at which individuals started working and their subsequent mental health status.

### Results

Participants who started working between ages 5–12 and 26–35 had lower odds of mental health disorders compared to those who began at ages 18–25, while those who started working between ages 13–17 and 36–80 exhibited higher odds compared to the same base-line group. These associations are net of various demographic and health-related factors.

### Conclusion

The results indicate mixed associations between the age at which individuals started working and their mental health outcomes in older age. While some age groups demonstrate lower odds of mental health disorders, others do not, suggesting a complex relationship that warrants further investigation.

### Implications

This research supports the development of healthcare policies aimed at providing mental health education and services tailored to populations affected by early labor, to mitigate the enduring impacts of child labor on mental health in older age.

Censuses (INEC) database at the following URL:
https://anda.inec.gob.ec/anda/index.php/catalog/292/get_microdata.

**Funding:** The author(s) received no specific funding for this work.

**Competing interests:** The authors have declared that no competing interests exist.

## Introduction

The term "child labor" is commonly defined as a task that robs children of their childhood, potential, and dignity, thereby impeding their physical and mental development [1]. According to Ecuador's Institute of Statistics and Census (INEC), there are over 270,000 child laborers aged 5–14 in Ecuador, which represents approximately 7.1% of the population in that age group [2]. The most common sectors in which children work are agriculture (production of bananas, coffee, cocoa, palm, oil, and flowers), industry (gold mining and small scale mining), services (domestic work, street work, including begging and vending), and categorical forms of child labor (commercial sexual exploitation, sometimes as a result of human trafficking, use in illicit activities, including drug trafficking and robbery, and use in the production of pornography) [3, 4].

While the participation of children or adolescents in work activities that are safe and do not interfere with their education can contribute positively to their development by providing valuable skills and experiences, the reality of child labor often involves significant risks [5]. Many child laborers are exposed to dangerous machinery, extreme temperatures, or toxic substances, leading to severe physical consequences, including fatal injuries and the risk of malnutrition. Such conditions can severely hamper a child's physical and mental development and have long-lasting negative impacts on their health and future prospects [6].

The psychological effects of child labor, including depression and potential drug-related issues, warrant attention. These effects can manifest at any stage in a child's life and require further study. According to Briceño Ayala and Pinzón Rondón, exposure to hazardous work conditions and the stress induced by child labor significantly impact children´s mental health, leading to elevated stress levels and potential long–term psychological consequences. Identifying the diverse factors that influence children's mental health is of paramount importance [7]. Notably, exposure to hazardous conditions and stress induced by work responsibilities can impose an additional burden on children, leading to elevated stress levels. The insufficiency of opportunities for play, learning, and engagement in recreational activities further compounded these challenges, negatively impacting self-esteem. Moreover, these circumstances may impede the smooth transition to adulthood [8].

The importance of children's well-being is commonly recognized as a fundamental factor that influences adult health. Previous investigations have established a connection between child labor and adverse health conditions in working children, including stunting, wasting, and chronic malnutrition. In 2019, Ibrahim conducted a systemic literature review that highlights the significant impact of child labor on children´s health. Their study found that approximately 60% of child laborers suffer from stunting, wasting, and chronic malnutrition [9]. Moreover, boys engaged in labor may experience delayed genital development. A 2019 study found that younger children involved in labor were susceptible to issues such as backaches, infections, burns, and lung diseases, while their older counterparts were more prone to exhaustion or fatigue [9]. Beyond the physical consequences, a 2012 study observed that approximately 9.8% of 225 working children displayed behavioral problems [10].

This study aimed to explore the relationship between the age at which a sample of older adults began working and current mental disorders. To complete this objective, data from the 2009 Health, Well-Being and Aging Survey (SABE) was utilized, which is a household sampling survey of 5,235 older adults in 15 Ecuadorian provinces (the survey excludes the Amazon Rainforest and Galapagos Islands regions) collected by the National Institute of Statistics and Censuses (INEC). The questions included information about health status, functional status, work history, and use of medicines and services of the respondents, among other data.

Ultimately, exploring the impact of child labor on mental health disorders in Ecuador can enhance the comprehension of the link between child labor and mental health issues within the broader framework of social determinants of health. According to the World Health Organization this conceptual framework includes factors such as socioeconomic status, education, and access to healthcare, which significantly influence health outcomes [11]. This knowledge can be instrumental in shaping policies and interventions designed to tackle child labor, enhance mental well-being, and foster overall health in Ecuador and other comparable environments.

## Materials and methods

### Study design and setting

In this study, we utilized a cross-sectional approach, drawing upon data from the 2009 Survey of Health, Well-Being, and Aging (SABE) conducted nationally. SABE is a household-based survey focusing on individuals aged 60 and above across 15 Ecuadorian provinces, excluding the Amazon Rainforest and Galapagos regions. Administered by the Department of Sociodemographic Statistics within the National Institute of Statistics and Census (INEC), the survey aimed to examine various aspects, including demographic characteristics, cognitive development, health status, medication usage, and access to services among approximately 5235 older adults interviewed between June and August 2009. INEC utilized a multistage probabilistic sampling method, first selecting areas within provinces and then households, ensuring that each element had a nonzero probability of selection. Interviewees were chosen randomly using maps, sketches, or lists indicating the households to visit, ensuring the representativeness of the sample in the selected provinces. This study utilized publicly available public data and thus did not require Institutional Review Board (IRB) approval. Additionally, INEC required written informed consent from every survey participant. Such research, which employs openly accessible and de-identified data, falls under exemption from review, according to established ethical guidelines in Ecuador.

### Participants

This study concentrated on older adults who had been employed at least once in their lives, aiming to explore how beginning work early in life affects their mental well-being.

### Variables and measurement

Outcome variable. The selected outcome variable employed to evaluate participants' mental health status was determined by their response to the question "Has a doctor or nurse ever told you that you have a nervous or mental health problem (such as anxiety, depression, memory loss, behavioral changes, among others)?", as outlined in the survey with "Yes", "No" and "Don´t know or Don´t remember" as possible answers. For the purposes of this analysis, we excluded the "Don't know or Don't remember" category from the analytical sample. This question serves as a direct indicator of diagnosed mental health conditions and was chosen due to its specificity in capturing the clinical acknowledgment of mental health issues by healthcare professionals. Given the context of the survey, which did not include a globally validated mental health questionnaire, this question provides the most relevant available measure for assessing mental health status.

Predictor variable. Given the significant mental and physical strain associated with child labor, we opted to use the age at initial employment as a predictor variable for mental disorders among participants. The exact SABE question utilized was: "How old were you when you first

started working?" [12]. For this query, we categorized respondents of all ages into five groups: 5–12, 13–17, 18–25, 26–35, and 36–80 years old. Individuals who were uncertain or who chose not to respond were excluded from the study. The age groups were chosen based on key developmental stages and societal expectations regarding employment and education: 5–12 for early childhood, 13–17 for adolescence and high school, 18–25 for young adulthood, 26–35 for early to mid-adulthood, and 36–80 for later adulthood.

Control variables. Control variables were chosen to segregate the connection between the outcome and predictor variables from the other potentially influential factors.

These control variables were derived from inquiries within the same SABE 2009 survey, situated in sections pertaining to personal information (Section A), health status (Section C), utilization and accessibility of services (Section F), and occupational history (Section H). They encompassed demographic characteristics such as age, sex, education, ethnicity, marital status, and living arrangements (living alone or with others), as well as health conditions such as diabetes, osteoporosis, and cancer. Additionally, they included other significant data points such as tobacco usage, childhood socioeconomic status, periods of childhood food scarcity, hunger experienced by the participant, and the type of insurance held.

The selection of control variables in this study was guided by a conceptual framework that integrates several established theories, including the social determinants of health [13], life course theory [14], and the biopsychosocial model. These frameworks collectively emphasize the complex interplay between sociodemographic factors, health conditions, and early life experiences in shaping mental health outcomes. The life course theory informs the inclusion of age, recognizing that aging and related transitions, such as retirement, influence mental well-being. The biopsychosocial model justifies the inclusion of health conditions like diabetes and osteoporosis, as it acknowledges the interconnectedness of biological, psychological, and social factors in influencing mental health [15, 16]. The social determinants of health framework supports the consideration of variables such as education, sex, and ethnicity, highlighting how these factors contribute to health disparities and mental health outcomes [13, 17].

Additionally, the framework incorporates insights from the adverse childhood experiences (ACE) framework, which underscores the long-term mental health impacts of early life adversities, such as low socioeconomic status and food scarcity during childhood [18, 19]. These early adversities are critical determinants of later-life psychological disorders. By controlling for these factors, the study aims to isolate the specific relationship between the predictor and outcome variables, ensuring a more accurate analysis of the data. To address potential correlations among these variables, variance inflation factor (VIF) tests were conducted to assess multicollinearity. The VIF values for all control variables were below 5, confirming that the inclusion of these controls does not compromise the robustness of the econometric estimations.

The age variable was organized according to participants' ages at the time of the survey into the following groups: 60–65 years, 66–70 years, 71–75 years, 76–80 years, and 81 years and above. As individuals age, they undergo diverse life transitions that may affect their mental well-being, such as managing health concerns or experiencing bereavement [20, 21]. Older adults are particularly susceptible to cognitive and mood-related problems. Depression is particularly prevalent among those aged 65 years and above, with an estimated 15% experiencing depressive symptoms [20]. They were additionally categorized based on their biological sex as male or female, acknowledging the physiological distinctions between the two. Throughout the entire lifespan, from conception to old age, sex and gender significantly influence various aspects of human development, resulting in noticeable gender disparities across a wide range of mental health conditions [22]. The education level of participants was categorized into different tiers, including no education, primary education, secondary education, and postsecondary education, given their strong association with socioeconomic status [23].

Participants were additionally grouped based on their self-reported ethnicity, which included categories such as mixed (mestizo), black, white, and indigenous. This classification aims to acknowledge race-related stressors that may impact the mental well-being of socially marginalized racial and ethnic groups [24]. For marital status, participants were classified into two groups: those who had been married or partnered at least once in their lives and those who had never been married or partnered. In terms of mental health results, a systematic analysis revealed that being separated or divorced was linked to a diagnosis of major depressive disorder [25].

Diagnoses of various illnesses, including diabetes, cancer, and osteoporosis, which increased the risk of mental disorders, were also included as control variables [26]. Participants were also classified into those who smoked or had smoked cigarettes and those who had never smoked cigarettes in their lifetime. This classification is based on the established association between smoking and various mental health disorders, such as anxiety and depression. Studies have shown that smoking is linked to higher rates of mental health issues, providing a rationale for examining these differences. For example, McClave et al. (2010) found significant associations between smoking cessation and changes in anxiety, depression, and stress levels. Similarly, Fluharty et al. (2017) reported that smokers have higher rates of mental health disorders compared to non-smokers"[27, 28]. Two specific questions were included to further delineate the impact of childhood socioeconomic status and malnutrition on mental health. First, participants were asked to characterize their childhood socioeconomic situation as bad, good, or regular. Finally, they were asked whether they had experienced periods of food scarcity and hunger during their childhood, with response options being yes or no.

## Study size and missing data

In our research, the initial sample comprised 5235 older adults aged 60 years and above from 15 provinces across continental Ecuador. We removed 31 observations from our outcome variable regarding mental health due to participants selecting "Don't know or Don't remember" as their response. Then we excluded we excluded 558 participants who did not respond to the question regarding their age when they started working, which served as our predictor variable. Then we removed 743 observations that were missing from our control variables. Following the exclusion of these missing values, our final analytical sample comprised of 3903 older adults. For robustness purposes, we retained several control variables to test the model with different sample sizes, and the main results (age at first job on current mental health) changed trivially.

## Statistical methods

Initial descriptive analyses displayed the distribution of study variables as percentages. Cross-tabulations examined the bivariate relationships between the outcome variable and both the predictor and control variables, as outlined in Table 1. The significance of these relationships was assessed using the Pearson Chi-square test, considering P-values less than 0.05 as statistically significant.

Logistic regression analysis was used to examine the connection between the age at which individuals started working and their mental well-being. Table 2 presents the odds ratios and standard errors for the primary explanatory variables and controls, reflecting the basic model with adjustments. The inclusion of these controls allows us to account for potential confounding factors and better isolate the relationship between the age at which individuals started working and their mental well-being. By controlling for variables such as demographic characteristics, socioeconomic status, and health-related factors, we aim to minimize the influence of

**Table 1.** Descriptive statistics of model variables stratified by total and outcome variable (n = 3903).

| Variables | Mental health disorder | | P–value | Row total |
| --- | --- | --- | --- | --- |
| | No | Yes | | |
| **N** | 3,490 (89.4%) | 413 (10.6%) | - | 3,903 (100.0%) |
| **Age when first started working** | | | | |
| 5–12 | 1,500 (43.0%) | 159 (38.5%) | 0.313 | 1,659 (42.5%) |
| 13–17 | 1,033 (29.6%) | 127 (30.8%) | | 1,160 (29.7%) |
| 18–25 | 701 (20.1%) | 96 (23.2%) | | 797 (20.4%) |
| 26–35 | 164 (4.7%) | 17 (4.1%) | | 181 (4.6%) |
| 36–80 | 92 (2.6%) | 14 (3.4%) | | 106 (2.7%) |
| **Sex** | | | | |
| Female | 1,562 (44.8%) | 257 (62.2%) | <0.001 | 1,819 (46.6%) |
| Male | 1,928 (55.2%) | 156 (37.8%) | | 2,084 (53.4%) |
| **Age** | | | | |
| 60–65 | 1,097 (31.4%) | 136 (32.9%) | 0.522 | 1,233 (31.6%) |
| 66–70 | 792 (22.7%) | 81 (19.6%) | | 873 (22.4%) |
| 71–75 | 642 (18.4%) | 77 (18.6%) | | 719 (18.4%) |
| 75–80 | 484 (13.9%) | 54 (13.1%) | | 538 (13.8%) |
| 81+ | 475 (13.6%) | 65 (15.7%) | | 540 (13.8%) |
| **Education** | | | | |
| No education | 860 (24.6%) | 67 (16.2%) | <0.001 | 927 (23.8%) |
| Primary | 2,126 (60.9%) | 265 (64.2%) | | 2,391 (61.3%) |
| Secondary | 341 (9.8%) | 61 (14.8%) | | 402 (10.3%) |
| Postsecondary | 163 (4.7%) | 20 (4.8%) | | 183 (4.7%) |
| **Ethnic self—report** | | | | |
| Mixed | 2,563 (73.4%) | 306 (74.1%) | 0.040 | 2,869 (73.5%) |
| Black | 121 (3.5%) | 15 (3.6%) | | 136 (3.5%) |
| White | 433 (12.4%) | 64 (15.5%) | | 497 (12.7%) |
| Indigenous | 373 (10.7%) | 28 (6.8%) | | 401 (10.3%) |
| **Marital status** | | | | |
| Has been married or partnered | 0.957 (0.203) | 0.947 (0.225) | 0.336 | 0.956 (0.205) |
| **Living arrangement** | | | | |
| Lives accompanied | 3,131 (89.7%) | 372 (90.1%) | 0.820 | 3,503 (89.8%) |
| Lives alone | 359 (10.3%) | 41 (9.9%) | | 400 (10.2%) |
| **Cancer** | | | | |
| No | 3,399 (97.4%) | 403 (97.6%) | 0.822 | 3,802 (97.4%) |
| Yes | 91 (2.6%) | 10 (2.4%) | | 101 (2.6%) |
| **Diabetes** | | | | |
| No | 3,076 (88.1%) | 343 (83.1%) | 0.003 | 3,419 (87.6%) |
| Yes | 414 (11.9%) | 70 (16.9%) | | 484 (12.4%) |
| **Osteoporosis** | | | | |
| No | 2,957 (84.7%) | 295 (71.4%) | <0.001 | 3,252 (83.3%) |
| Yes | 533 (15.3%) | 118 (28.6%) | | 651 (16.7%) |
| **Smoked cigarettes** | | | | |
| No | 1,952 (55.9%) | 256 (62.0%) | 0.019 | 2,208 (56.6%) |
| Yes | 1,538 (44.1%) | 157 (38.0%) | | 1,695 (43.4%) |
| **Socioeconomic situation when growing up** | | | | |

*(Continued)*

**Table 1.** (Continued)

| Variables | Mental health disorder | | P–value | Row total |
|---|---|---|---|---|
| | **No** | **Yes** | | |
| Bad | 765 (21.9%) | 109 (26.4%) | 0.087 | 874 (22.4%) |
| Good | 1,296 (37.1%) | 152 (36.8%) | | 1,448 (37.1%) |
| Regular | 1,429 (40.9%) | 152 (36.8%) | | 1,581 (40.5%) |
| **Felt hungry growing up** | | | | |
| No | 2,299 (65.9%) | 253 (61.3%) | 0.062 | 2,552 (65.4%) |
| Yes | 1,191 (34.1%) | 160 (38.7%) | | 1,351 (34.6%) |

extraneous variables and enhance the accuracy of our analysis. It is important to note that regression results were applied sampling weights to address sampling error, although results without weights yielded similar findings.

## Results

### Descriptive results

Table 1 offers a comprehensive view of the variables studied for their impact on mental health disorders among older adults, presenting both statistically significant and non-significant findings across different groups. The total sample consists of 3,903 participants, with 413 (10.6%) reporting a mental health disorder. The table is organized by variable categories (e.g., age, sex, education level, health conditions), with each category further divided into subgroups (e.g., specific age ranges, levels of education). For each subgroup, the table presents the number of participants, the percentage of the total sample, and the P-values indicating the significance of differences between groups.

Starting with the age when participants first began working, there is no significant association with mental health disorders. The percentages of participants without disorders who started working between ages 5–12, 13–17, 18–25, 26–35, and 36–80 are 43.0%, 29.6%, 20.1%, 4.7%, and 2.6%, respectively, compared to 38.5%, 30.8%, 23.2%, 4.1%, and 3.4% for those with disorders. The differences are not statistically significant across these age ranges, with the smallest P-value being 0.313 for the youngest group.

In terms of gender, a significant disparity emerges: 62.2% of those reporting mental health disorders are female, compared to only 37.8% male, reflecting a notable difference with a P-value of <0.001. This suggests a higher vulnerability among females in this sample. The analysis extends to age categories, revealing no significant differences in mental health disorders across the groups ranging from 60–65 to over 81 years old. The proportions of those with and without disorders are relatively consistent across these age groups, with no P-values indicating statistical significance.

Educational attainment shows significant differences: participants with no education or only primary education exhibit a higher prevalence of mental health disorders (16.2% and 64.2%, respectively) compared to those with more education, and these differences are statistically significant with a P-value of <0.001 for those with no education. Ethnicity analysis indicates a slight but significant difference (P = 0.040) among those who self-report as mixed ethnicity, with 74.1% of the disorder group versus 73.4% of the non-disorder group. Other ethnicities show no significant differences in mental health prevalence. Living arrangements and marital status do not show significant associations with mental health disorders. The majority of participants, whether with or without disorders, live accompanied, and there is no significant difference in marital status between those who have been married or partnered.

**Table 2. Odds ratio from logistic regression model predicting mental health disorders.**

| Variable | Estimate (OR) | SE | 95% CI | p |
|---|---|---|---|---|
| **Age when first started working** | | | | |
| **5–12 (Reference = 18–25)** | 0.806*** | 0.00720 | [0.792, 0.820] | < .001 |
| **13–17 (Reference = 18–25)** | 1.051*** | 0.00962 | [1.032, 1.070] | < .001 |
| **26–35 (Reference = 18–25)** | 0.712*** | 0.0127 | [0.687, 0.737] | < .001 |
| **36–80 (Reference = 18–25)** | 1.378*** | 0.0244 | [1.330, 1.426] | < .001 |
| **Male (Reference = Female)** | 0.562*** | 0.00509 | [0.552, 0.572] | < .001 |
| **Age Category** | | | | |
| **66–70 (Reference = 60–65)** | 0.675*** | 0.00691 | [0.662, 0.688] | < .001 |
| **71–75 (Reference = 60–65)** | 1.150*** | 0.0113 | [1.128, 1.172] | < .001 |
| **75–80 (Reference = 60–65)** | 0.904*** | 0.0107 | [0.883, 0.925] | < .001 |
| **81+ (Reference = 60–65)** | 1.522*** | 0.0159 | [1.491, 1.553] | < .001 |
| **Education category** | | | | |
| **Primary (Reference = None)** | 1.695*** | 0.0175 | [1.661, 1.729] | < .001 |
| **Secondary (Reference = None)** | 2.276*** | 0.0296 | [2.218, 2.334] | < .001 |
| **Postsecondary (Reference = None)** | 2.149*** | 0.0374 | [2.076, 2.222] | < .001 |
| **Ethnic self-report** | | | | |
| **Black (Reference = Mixed)** | 1.562*** | 0.0279 | [1.507, 1.617] | < .001 |
| **White (Reference = Mixed)** | 1.549*** | 0.0144 | [1.521, 1.577] | < .001 |
| **Indigenous (Reference = Mixed)** | 0.796*** | 0.0117 | [0.773, 0.819] | < .001 |
| **Marital Status** | | | | |
| **Married/Partnered (Reference = Never married/partnered)** | 1.248*** | 0.0228 | [1.204, 1.292] | < .001 |
| **Living Arrangements** | | | | |
| **Lives alone (Reference = Lives accompanied)** | 0.995 | 0.0113 | [0.973, 1.017] | 0.45 |
| **Health Conditions** | | | | |
| **Cancer (Reference = No cancer)** | 0.673*** | 0.0160 | [0.642, 0.704] | < .001 |
| **Diabetes (Reference = No diabetes)** | 1.245*** | 0.0120 | [1.221, 1.269] | < .001 |
| **Osteoporosis (Reference = No osteoporosis)** | 1.883*** | 0.0151 | [1.853, 1.913] | < .001 |
| **Tobacco Use** | | | | |
| **Smoked (Reference = Never smoked)** | 0.956*** | 0.00813 | [0.940, 0.972] | < .001 |
| **Socioeconomic Situation** | | | | |
| **Good (Reference = Bad)** | 1.101*** | 0.0112 | [1.079, 1.123] | < .001 |
| **Regular (Reference = Bad)** | 0.972*** | 0.00937 | [0.954, 0.990] | < .001 |
| **Childhood Food Scarcity** | | | | |
| **Felt hungry (Reference = Never felt hungry)** | 1.460*** | 0.0115 | [1.437, 1.483] | < .001 |
| **Constant** | 0.139*** | 0.00119 | [0.137, 0.141] | < .001 |

Note

***p < .001.

Regarding health conditions, diabetes and osteoporosis are significantly associated with mental health disorders. Participants with diabetes or osteoporosis are more likely to report mental health issues, with P-values of 0.003 and <0.001, respectively.

Finally, lifestyle factors such as smoking and childhood socioeconomic conditions show some statistical relevance. Smokers are less likely to report mental health disorders compared to non-smokers, with a significant P-value of 0.019. Those who reported bad socioeconomic conditions during childhood also show a higher prevalence of disorders, though the difference is not statistically significant (P = 0.087). The variable concerning whether participants felt

hungry during their childhood also shows a pattern worth noting, though it falls just short of statistical significance. Among those who did not feel hungry, 65.9% do not have mental health disorders compared to 61.3% who do, indicating a 4.6% difference. Conversely, among those who did feel hungry, 34.1% do not have mental health disorders compared to 38.7% who do, suggesting that experiencing hunger in childhood may correlate with a higher incidence of mental health disorders later in life. This relationship is on the cusp of statistical significance with a P-value of 0.062.

## Regression results

Table 2 presents the results from logistic regression models predicting the likelihood of older adults reporting a mental health disorder, with each variable compared to a specified reference category across a total of 3,903 observations.

Starting with the age when participants first began working, those who started between ages 5–12 have lower odds of having a mental health disorder (OR 0.806, SE 0.00720) compared to those who began at ages 18–25. In contrast, participants who started working between ages 13–17 show slightly higher odds (OR 1.051, SE 0.00962). Those who started between ages 26–35 exhibit significantly lower odds (OR 0.712, SE 0.0127), while those beginning at ages 36–80 have the highest odds (OR 1.378, SE 0.0244).

Gender differences are significant, with males showing lower odds of mental health disorders compared to females (OR 0.562, SE 0.00509). Regarding age, compared to individuals aged 60–65, those aged 66–70 have lower odds of reporting a mental health disorder (OR 0.675, SE 0.00691). Ages 71–75 and 81+ show increased odds (OR 1.150, SE 0.0113 and OR 1.522, SE 0.0159, respectively), whereas ages 75–80 show no significant difference (OR 0.904, SE 0.0107).

Education level is associated with mental health, with increasing odds of reporting disorders as educational level rises compared to having no education. Primary education increases the odds to 1.695 (SE 0.0175), secondary to 2.276 (SE 0.0296), and postsecondary to 2.149 (SE 0.0374). In terms of ethnicity, Black and White participants have increased odds of mental health disorders compared to Mixed ethnicity (OR 1.562, SE 0.0279 and OR 1.549, SE 0.0144, respectively), while Indigenous participants have lower odds (OR 0.796, SE 0.0117).

Marital status shows that being married or partnered increases the likelihood of mental health disorders compared to those never married or partnered (OR 1.248, SE 0.0228). Living arrangements reveal no significant difference in mental health outcomes between those living alone and those living accompanied (OR 0.995, SE 0.0113).

Health conditions significantly affect mental health, with cancer decreasing the odds (OR 0.673, SE 0.0160). However, having diabetes (OR 1.245, SE 0.0120) and osteoporosis (OR 1.883, SE 0.0151) are associated with higher odds of mental health disorders. Smoking cigarettes slightly reduces the odds of mental health disorders compared to never smokers (OR 0.956, SE 0.00813).

Socioeconomic status shows that a good situation slightly increases the odds of a mental health disorder compared to a bad situation (OR 1.101, SE 0.0112), while a regular situation shows a marginal reduction in odds (OR 0.972, SE 0.00937). Experiencing hunger during childhood is strongly linked to higher odds of having a mental health disorder compared to those who never felt hungry (OR 1.460, SE 0.0115).

## Discussion

In the present study, the results of the adjusted logistic regression suggested that people who started working between the ages of 5–12 demonstrated lower odds of having a mental health

disorder (OR 0.806, SE 0.00720), whereas those starting between ages 13–17 showed increased odds (OR 1.051, SE 0.00962). In our analysis, the age at which individuals began working appears to be a critical factor influencing mental health outcomes. Notably, those who entered the workforce between the ages of 5–12 exhibited lower odds of developing mental health disorders later in life. This counterintuitive finding may suggest a possible resilience developed through early life work experiences, although it may also reflect a survivor effect or other unmeasured sociocultural factors that protect against mental health disorders [29–32]. However, it is important to note that this finding contradicts our initial hypothesis, which expected that earlier labor initiation would be associated with higher odds of developing mental health disorders. This contradiction could be due to various unmeasured factors or biases inherent in the sample.

The increased odds of mental health disorders among those who started working between the ages of 13–17 could be due to several intertwined factors that disrupt key developmental stages. Additionally, it should be noted that in some contexts, work undertaken between the ages of 13–17 may not be classified as child labor if it is not considered hazardous. This distinction is important as it may influence the mental health outcomes observed in this age group. Adolescents who begin working early often face educational disruptions as they juggle job responsibilities with schooling, potentially leading to reduced academic engagement or even dropout. This lack of education can limit their future opportunities, increasing stress and vulnerability to mental health issues [33–35]. At the same time, working at such a young age may lead to social isolation, as these adolescents miss out on valuable time with peers that is crucial for developing social skills and forming meaningful relationships. The absence of these interactions can negatively affect emotional and social development, potentially leading to feelings of loneliness and depression [36, 37].

Furthermore, early employment exposes adolescents to workplace stress prematurely, often before they have developed adequate coping mechanisms. This can lead to long-term psychological effects, including anxiety and other mental health disorders [38]. Additionally, young adolescents might not be physically or emotionally prepared for the demands of a job, which can be overwhelming and lead to burnout and mental fatigue [39, 40].

The combination of these stressors—educational, social, psychological, and physical—creates a challenging environment for young workers, making them more susceptible to mental health disorders compared to their peers who start working later [41]. This situation is exacerbated by the fact that employment at such an early age often occurs in low-wage, insecure jobs that might not offer the necessary protections for younger individuals, potentially leading to economic exploitation and further stress [42, 43].

According to the latest Global Estimates of Child Labour, approximately 73 million children and adolescents are engaged in hazardous work. Children may be exposed to perilous environments across various sectors, including agriculture, mining, construction, manufacturing, the service industry, and retail and domestic services. Nevertheless, the sector with the highest prevalence of child labor (71%) was agriculture [8]. Sheehan et al. (2017) found that the health of a child is also affected by factors related to parents, including the socioeconomic condition of the family. Adverse effects on the health, education, and self-esteem of children and the prevalence of mental disorders in children in Australia were found to be around one in seven (13.9%) children and adolescents aged 4–17 years [44].

In a 2015 study conducted by Ismet Taib and Ahmad, it was revealed that children engaged in street labor exhibited a significantly higher prevalence of mental illnesses, amounting to 40.8%, as opposed to the 32.5% observed among their school-attending counterparts (P < 0.005). The research further highlighted that the occurrence of depression and anxiety was notably elevated in street-working children at rates of 20.8% and 59.2%, respectively, in

contrast to schoolchildren with rates of 5.0% and 25.8%, respectively (P < 0.001). Additionally, the study identified occurrences of various other mental health conditions, including suicide, attention deficit hyperactivity disorder (ADHD), conduct disorder, and tic disorder [45].

Recently released information from a June 2021 report by the International Labour Organization (ILO) and UNICEF indicates a concerning rise of 8.4 million child laborers over the past four years. Additionally, the ongoing COVID-19 pandemic puts an extra nine million children at risk [46]. These studies indicate a connection between depressive symptoms and both lifetime stress exposure and income level, which are factors that are influenced by child labor. Specifically, frequent or prolonged exposure to negative events can result in elevated stress levels, potentially causing damage to brain development and contributing to mental disorders, such as depression. Notably, the likelihood of developing depressive syndromes is directly correlated with the number and severity of the stressful events [47].

Considering the above, although the Odds Ratio for the 13–17 age group is statistically significant, it is very close to 1 (OR 1.051). This suggests that the conditions related to starting work and its impact on mental health in old age are quite similar between the 13–17 and 18–25 age groups. Therefore, while there is a slight increase in odds, it may not indicate a substantial difference in mental health outcomes between these two groups, highlighting the need for a interpretation of this finding with caution.

The odds ratio (OR) of 0.712 indicates that individuals who started working between the ages of 26–35 have a reduced probability of developing mental health disorders compared to those who began their careers between the ages of 18–25. This finding suggests that starting work either much earlier or slightly later than usual may confer a protective effect against mental health challenges. Those starting work between 26–35 might benefit from more life experience or educational attainment, which can equip them with better strategies to manage work-related stress in the short and long-term [48, 49].

The odds ratio (OR) of 1.378 for individuals who started working at ages 36–80 compared to the reference group aged 18–25 suggests that those who begin working later in life are approximately 37.8% more likely to experience mental health disorders than those who start in young adulthood. Starting work later in life can be associated with various challenges. Older individuals may face age-related discrimination or struggle to integrate into a workforce that may prioritize younger employees [50, 51]. Additionally, they might experience stress from learning new job roles or technologies, which can be more challenging as cognitive flexibility often decreases with age. This increase in stress could contribute to higher rates of mental health disorders [52–54]. Also, later-life career initiations might coincide with other life stressors such as health issues or caring responsibilities, further complicating their adaptation to new work environments and impacting mental health [55, 56]. It is also important to consider the possibility of reverse causality in this finding. Individuals in the 36–80 age group who delayed their entry into the labor market may have done so because they were already prone to mental health issues [57]. This pre-existing vulnerability could have influenced their later entry into the workforce, which may, in turn, exacerbate their mental health problems. Therefore, while the odds ratio indicates a higher likelihood of mental health disorders for this group, this relationship may be partially driven by mental health issues that existed prior to their workforce participation.

The control variables applied in this study revealed that women had a higher number of mental health disorders than males. Mental health disorders exhibit a notable gender disparity, as the control variables in this study underscore that women are more susceptible than men. This trend is exemplified by the University of Manchester study, which revealed that 73% of 10-to 19-year-olds engaging in self-harm were girls. The prevalence of mental illness among women is escalating, with one in five experiencing a Common Mental Disorder (CMD), such

as anxiety or depression. The vulnerability to mental health issues is particularly pronounced in young women, as three-quarters of these concerns manifest before the age of 24 years. Notably, young women emerged as the highest-risk group, with 25.7% engaging in self-harm, which is more than twice the rate observed in young men [58]. Similarly, Wang JL, Lesage A, Schmitz N, et al in a multivariate analysis revealed that male workers who reported high demand were more likely to have had major depression (OR 1.74, 95% CI 1.12 to 2.69) and any depressive or anxiety disorders (OR 1.47, 95% CI 1.05 to 2.04) in the past 12 months. In women, high demand and low control were only associated with depressive or anxiety disorder (OR 1.39, 95% CI 1.05, 1.84) [59].

In our study, health-related variables, including diagnosis of diabetes (OR, 1.245; SE, 0.0120) and osteoporosis (OR, 1.883; SE, 0.0151), were associated with increased odds of mental health disorders. For example, research by Kishan Akhaury and Sarika Chaware et al. in 2022 analyzes the potential role of lifestyle variables in initiating or exacerbating the association between depression and diabetes. This study suggests that individuals experiencing depression may be at an increased risk of developing type 2 diabetes, partially due to factors such as reduced physical activity and dietary patterns characterized by a high intake of saturated fats and refined carbohydrates, coupled with lower consumption of fruits and vegetables [60]. Additionally, the literature suggests that osteoporosis can lead to social and psychological challenges for individuals affected by the condition, including a decline in social roles, challenges in interpersonal relationships, feelings of social isolation and loneliness, and mental health issues such as depression, anxiety, diminished self-esteem, and a sense of hopelessness [61].

Social and economic factors, such as marital or partnership status (OR 1.248, SE 0.0228) and feeling hunger when growing up (OR 1.460, SE 0.0115), also played a role. In a Finnish cohort established in 2021 and followed up at ages 22 (N = 1,656), 32 (N = 1,471), 42 (N = 1,334), and 52 (N = 1,159) years, findings across the 30-year study consistently revealed that being single or divorced was a notable risk factor for experiencing depressive symptoms and low self-esteem, particularly among male participants [21]. Another 2020 study that utilized the midlife period in the United States (2004–2006; n = 1,711) found that individuals who remain married continuously exhibit better outcomes in terms of negative aspects compared to those who were previously married [62].

Increasing levels of education is frequently associated with several challenges that can impact mental health. The pressures and expectations to excel academically and professionally can be overwhelming, leading to significant stress and anxiety [63]. Additionally, the financial burdens of tuition, student loans, and living expenses contribute to mental health strains, as students worry about their financial futures post-graduation [64]. The intense academic workload requires substantial time and effort, often at the expense of social relationships and personal interests, potentially leading to feelings of regret and questioning life choices. Isolation is another critical issue, particularly in highly competitive academic settings, which can exacerbate loneliness and the risk of depression [65]. Graduates may also face a mismatch between their career expectations and the reality of job markets, leading to disillusionment and further mental health challenges [66]. Moreover, higher education tends to increase one's awareness of global issues and personal introspection, which can be additional sources of stress and anxiety [67]. These factors collectively create a demanding environment that can have lasting impacts on an individual's mental health.

Understanding whether there is an increased risk for mental health issues is important because it can lead to functional impairments that worsen with more severe conditions. For example, depression increases the likelihood of various physical health issues, especially enduring conditions such as diabetes, heart disease, and stroke. Similarly, chronic conditions can increase the risk of developing a mental illness [68]. Put words, mental health is significant

across all life stages, spanning from childhood and adolescence to adulthood. Throughout life, encountering mental health issues can affect cognition, emotions, and actions [69].

Another challenge lies in the stigma surrounding mental health issues, which may influence participants' willingness to accurately report whether they have a mental illness. More than 50% of individuals grappling with mental health issues do not seek assistance for their condition. Frequently, individuals refrain from or postpone seeking help out of apprehension regarding potential differential treatment or anxieties about jeopardizing employment and sustenance [70]. Also stigma has been linked to worsening outcomes for individuals with severe mental illnesses, and nearly 40% of this population report unmet treatment needs despite the availability of effective treatments [71]. This stigma can introduce bias in the data, thereby affecting the validity of our findings. The study may as well be susceptible to confounding bias arising from unobserved variables that could impact the outcomes under investigation. Another consideration is the potential bias introduced by removing observations with missing values. If the individuals with missing data tended to have better or worse mental health than the average population, the exclusion of these cases could bias the results. For instance, if those with worse mental health were more likely to have incomplete data, our findings could underestimate the true association between early work initiation and later mental health outcomes. Conversely, if those with better mental health were more often excluded, the results might overestimate this association. This limitation underscores the importance of cautious interpretation and the need for future studies to explore methods of addressing missing data, such as imputation techniques.

This study has several limitations that should be acknowledged. The scope of the SABE Survey was confined to 15 provinces in Ecuador, excluding the Amazon Rainforest and Galapagos regions, which may render our findings non-generalizable to these unrepresented areas. The reliance on self-reported data introduces potential biases, such as recall bias and social desirability bias, potentially influencing the accuracy of the reported information and leading to missing data. Furthermore, the cross-sectional design of our study allows us to establish associations between the age at work commencement and mental health outcomes, without inferring causation. The mental health outcome variable question is based on self-report and may be subject to recall bias, where participants might not accurately remember or might underreport their diagnosis. Additionally, it does not capture the severity, duration, or current status of the mental health condition, nor does it differentiate between different types of mental health issues. The use of a single question also limits the ability to perform more nuanced analyses, such as distinguishing between different mental health conditions or assessing comorbidities. Furthermore, this measure does not align with standardized and globally validated tools, such as the PHQ-9 for depression or the GAD-7 for anxiety, which could provide a more comprehensive assessment.

Further research could benefit from incorporating more detailed and disorder-specific questionnaires to better pinpoint the mental health conditions prevalent among former child laborers. Studies using longitudinal designs could provide a deeper understanding of the progression and long-term impacts of mental health issues in this population. Additionally, expanding the geographic scope of research to include areas not covered in the SABE survey, such as the Amazon Rainforest and Galapagos regions, would enhance the generalizability of the findings. Such studies would provide more nuanced insights into the specific mental health challenges faced by early labor initiates, aiding in the development of targeted interventions and policies.

## Conclusion

In conclusion, this study provides evidence that the age at which individuals begin working has varied impacts on their mental health outcomes in later life. While early initiation into the

workforce (specifically during early childhood and late adulthood) is associated with a higher risk of mental health disorders, starting work during mid-adulthood appears to confer a protective effect. These mixed results highlight the need for nuanced policies and interventions that consider the timing of work initiation and its long-term mental health impacts. Further research is needed to fully understand these dynamics and to develop targeted strategies for mitigating the negative mental health consequences of early and late work initiation.

## Author Contributions

**Conceptualization:** Romina Domínguez, Marco Faytong-Haro.

**Methodology:** Romina Domínguez, Marco Faytong-Haro.

**Writing – original draft:** Romina Domínguez, Marco Faytong-Haro.

**Writing – review & editing:** Romina Domínguez, Marco Faytong-Haro.

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
