## [Decision Letter · Decision Letter 0]

9 Jul 2024

PONE-D-24-15715Impact of Early Work Start on Mental Health Outcomes in Older Adults: A Cross-Sectional Study from EcuadorPLOS ONE

Dear Dr. Faytong-Haro,

Thank you for submitting your manuscript to PLOS ONE. After careful consideration, we feel that it has merit but does not fully meet PLOS ONE’s publication criteria as it currently stands. Therefore, we invite you to submit a revised version of the manuscript that addresses the points raised during the review process.

This is an interesting paper presenting an interesting discussion and results. The paper, however, needs some additional work. Some general points: 1) It would be interesting to discuss the effects of removing observations with missing values. If these tended to be people with better or worse mental health than the average population, the results of the paper would be biased2) improve the discussion of the variables used in the model. It would be interesting to relate the variables with the literature and some discussion on the expected direction3) data presentation (tables) could be improved. 4) see addditional comments by the reviewers.  Please, see very detailed and careful comments made by the reviewers. 

We look forward to receiving your revised manuscript.

Kind regards,

Bernardo Lanza Queiroz, Ph.D

Academic Editor

PLOS ONE

Journal Requirements:

Reviewers' comments:

Reviewer's Responses to Questions

**Comments to the Author**

1. Is the manuscript technically sound, and do the data support the conclusions?

Reviewer #1: Partly

Reviewer #2: Yes

2. Has the statistical analysis been performed appropriately and rigorously? 

Reviewer #1: Yes

Reviewer #2: Yes

3. Have the authors made all data underlying the findings in their manuscript fully available?

Reviewer #1: Yes

Reviewer #2: Yes

4. Is the manuscript presented in an intelligible fashion and written in standard English?

Reviewer #1: Yes

Reviewer #2: Yes

5. Review Comments to the Author

Reviewer #1: This article provides an interesting case study on two very important variables, which are not often addressed in the literature: child labor and mental disorders in later life. It's well structured; the English grammar and text flow well, and the authors demonstrate a mastery of the literature and carefulness with the research methods.

I believe that the article could be accepted for publication, but it requires some changes in how authors interpret their results. They do not have strong evidence that child labor is associated with mental disorders. In the abstract, the authors try to provide compelling evidence in favor of their hypothesis, including in the conclusions. However, these claims seem overstated and are not well supported by the results.

Here are some suggestions for improving the paper:

1. Starting early in the abstract, the results appear quite mixed (some age groups show lower odds of mental health issues, while others do not). However, in the conclusion section, the authors assert confidently that the results support their hypothesis, which is not fully supported by the evidence.

2. "There are over 270,000 child laborers aged 5–14 in Ecuador." Although the number is high, and child labor is concerning, presenting this as a percentage would provide more informative context.

3. "The psychological effects of child labor, including depression and potential drug-related issues, warrant attention. These effects can manifest at any stage in a child's life and require further study." These statements should be supported by references from the literature. Are they extracted from [7]?

4. "Previous investigations have established a connection between child labor and adverse health conditions in working children, including stunting, wasting, and chronic malnutrition." Please provide citations for these findings.

5. "This study aimed to explore the relationship between the age at which a sample of older adults began working and mental disorders." It might be beneficial to clarify that you are focusing on current mental disorders.

6. "Ultimately, exploring the impact of child labor on mental health disorders in Ecuador can enhance the understanding of the link between child labor and mental health issues within the broader framework of social determinants of health." Please provide references for the conceptual framework used in this research, although the article's structure does not explicitly require this.

7. "Outcome variable. The selected outcome variable used to evaluate participants' mental health status was determined by their response to the question 'Has a doctor or nurse ever told you that you have a nervous or mental health problem (such as anxiety, depression, memory loss, behavioral changes, among others)?', as outlined in the survey." If permitted by the journal's guidelines, a more detailed description of this variable should be provided. What are its limitations? Is it based on a globally validated depression questionnaire? The current description seems insufficient.

8. For the outcome variable, please specify the categories (e.g., yes/no).

9. Regarding control variables, it would be important to explain how they are chosen based on a conceptual framework. Additionally, discuss potential correlations among control variables that might complicate the econometric estimation.

10. "As individuals age, they undergo diverse life transitions that may affect their mental well-being, such as managing health concerns or experiencing bereavement." Please provide references for these assertions.

11. "Participants were also classified into those who smoked or had smoked cigarettes and those who had never smoked cigarettes in their lifetime." Please provide the rationale for this classification and supporting references.

12. "We removed 31 observations from our outcome variable regarding mental health." Why were these observations removed?

13. "and the main results changed trivially." Could you clarify what you mean by "main results"? Specify the coefficient for which the sensitivity analysis was conducted.

14. "Starting with the age when participants first began working, those who started between ages 5–12 have lower odds of having a mental health disorder (OR 0.806, SE 0.00720)." It's important to highlight that this result contradicts your hypothesis. Also, note that in some contexts, working from ages 13–17 may not be considered child labor if the work is not hazardous.

15. The descriptive table is extensive and challenging to interpret. Consider replacing some statistics with graphs for clarity.

16. The format of the regression results table could be improved. Check APA style guidelines for presenting regression results (https://apastyle.apa.org/style-grammar-guidelines/tables-figures/sample-tables#regression).

17. "Notably, those who entered the workforce between ages 5–12 exhibited lower odds of developing mental health disorders later in life. This counterintuitive finding may suggest a possible resilience developed through early life work experiences, although it may also reflect a survivor effect or other unmeasured sociocultural factors that protect against mental health disorders [18–21]." Consider discussing how issues with recall bias regarding the variable "age at which the individual started working" might influence this result.

18. If space in the article is limited, focus less on discussing results for adults in their prime age. The primary concern of the paper is child labor.

Reviewer #2: • There is an interesting point about the results of the "age at which one started working" variable in the regression. Although statistically significant, the Odds Ratio for the 13-17 age group is very close to 1. Therefore, in interpreting this value, it can be said that the conditions relating age to starting work and mental health in old age are the same for the 13-17 and 18-25 age groups.

• For the Odds Ratio result of the 36-80 age group, the existence of reverse causality can be discussed. Perhaps these were people already prone to mental health problems, which is why they delayed their entry into the labor market for so long.

• It would be interesting to discuss the interaction effect between the variables age and age at which one started working. The greater or lesser propensity to have mental health problems may be affected not only by the age at which one started working but also by the period during which one started working - for example, a period of economic crisis. Depending on the period when one started working, there may also be changes in working conditions (assuming that the further back in time, the worse the working conditions were). This analysis would help to better understand what age and the age at which one started working mean for mental health.

• It would be interesting to discuss the effects of removing observations with missing values. If these tended to be people with better or worse mental health than the average population, the results of the paper would be biased.

6. PLOS authors have the option to publish the peer review history of their article (what does this mean?). If published, this will include your full peer review and any attached files.

Reviewer #1: **Yes: **Raquel Guimaraes

Reviewer #2: No

---

## [Author Response · Author response to Decision Letter 0]

26 Aug 2024

We have enclosed a Word document in which each reviewer and editor comment is addressed in a table format. This format is intended to enhance the reviewer's experience by making it easier to match each comment with our corresponding response. Please see the enclosed document for detailed responses.

---

## [Editor Report · Decision Letter 1]

1 Sep 2024

Impact of Early Work Start on Mental Health Outcomes in Older Adults: A Cross-Sectional Study from Ecuador

PONE-D-24-15715R1

Dear Dr. Faytong-Haro,

We’re pleased to inform you that your manuscript has been judged scientifically suitable for publication and will be formally accepted for publication once it meets all outstanding technical requirements.

Kind regards,

Bernardo Lanza Queiroz, Ph.D

Academic Editor

PLOS ONE
---

## [Editor Report · Acceptance letter]

7 Oct 2024

PONE-D-24-15715R1 

PLOS ONE

Dear Dr. Faytong-Haro, 

I'm pleased to inform you that your manuscript has been deemed suitable for publication in PLOS ONE. Congratulations! Your manuscript is now being handed over to our production team.

Kind regards, 

on behalf of

Dr. Bernardo Lanza Queiroz 

Academic Editor

PLOS ONE